# Quality Criteria and Method of Synthesis for Adversarial Attack-Resistant Classifiers

**Anastasia Gurina [1,2] and Vladimir Eliseev [1,2,*]**

1   JSC InfoTeCS, Otradnaya 2B building 1, Moscow 127273, Russia; asya.gurina001512@yandex.ru
2   Department of Control and Intellectual Technologies, National Research University "Moscow Power Engineering Institute", Krasnokazarmennaya 17, Moscow 111250, Russia
*   Correspondence: vlad-eliseev@mail.ru

**Abstract:** The actual problem of adversarial attacks on classifiers, mainly implemented using deep neural networks, is considered. This problem is analyzed with a generalization to the case of any classifiers synthesized by machine learning methods. The imperfection of generally accepted criteria for assessing the quality of classifiers, including those used to confirm the effectiveness of protection measures against adversarial attacks, is noted. The reason for the appearance of adversarial examples and other errors of classifiers based on machine learning is investigated. A method for modeling adversarial attacks with a demonstration of the main effects observed during the attack is proposed. It is noted that it is necessary to develop quality criteria for classifiers in terms of potential susceptibility to adversarial attacks. To assess resistance to adversarial attacks, it is proposed to use the multidimensional EDCAP criterion (Excess, Deficit, Coating, Approx, Pref). We also propose a method for synthesizing a new EnAE (Ensemble of Auto-Encoders) multiclass classifier based on an ensemble of quality-controlled one-class classifiers according to EDCAP criteria. The EnAE classification algorithm implements a hard voting approach and can detect anomalous inputs. The proposed criterion, synthesis method and classifier are tested on several data sets with a medium dimension of the feature space.

**Keywords:** adversarial attack modeling; adversarial examples; classification quality criteria; multiclass classification; multidimensional feature space; machine learning; deep neural networks; resistance to adversarial attacks



## 1. Introduction

Adversarial attacks are purposefully generated examples of input data that cause errors in classifiers based on machine learning methods. Usually, adversarial examples are defined through the minimum deviation from correctly classified examples [1]. In general, however, the closeness of examples is of little value if the classifier unpredictably makes a decision different from what a person in a similar situation would make.

Considering the widespread use of machine learning methods and classifiers based on those methods, there are many examples of the use of this feature by attackers for targeted attacks on artificial intelligence systems. The most vulnerable are intelligent systems for recognizing and classifying images and voices, but attacks on classifiers of network traffic, files, events of various kinds, and text are also possible. To attack such systems, attackers affect the input data by distorting the image, introducing noise into network traffic, modifying files, generating distracting events, and modifying text. Examples of real attacks are widely known, including an attack on a voice assistant—buying goods on Amazon Alexa [2]—and bypassing anti-spam filters [3]. It is known that some classifiers already used in production are susceptible to adversarial attacks, including computer vision systems [4–6], face recognition systems [7], and others [8]. These and other precedents of incorrect operation of artificial neural networks do not allow one to fully rely on this tool when solving problems with a high cost of error. These systems include the autopilot of

Tesla cars. The errors of the classifier built into the autopilot became resonant, as did the consequences [9]. The adversarial vulnerability of the computer vision system used in Tesla was previously identified by researchers [10–12].

The unpredictable behavior of machine learning algorithms makes the issue of protecting classifiers from adversarial attacks very relevant. At the same time, it cannot be said that the problem of protection against adversarial attacks has been solved or even well studied. Most of the proposed methods are heuristic and cannot guarantee the efficiency of solving the classification problem using machine learning methods in the general case. Moreover, methods for assessing the degree of vulnerability of a trained classifier to adversarial attacks have not been developed that would allow classifiers to be compared with each other.

In this paper, we investigate the reasons for the vulnerability of classifiers to adversarial attacks, and also propose a new multidimensional EDCAP quality criterion that allows us to quantify the susceptibility of a classifier to adversarial attacks. The generalized classifier model explains the main effects observed by researchers in experiments. Among these effects are the ability to build adversarial attacks in white- and black-box modes, as well as the susceptibility to adversarial attacks of machine learning models of various natures: neural networks, SVMs, decision trees, etc. A new method for synthesizing a multiclass classifier based on an ensemble of single-class neural network auto-encoders is introduced, allowing the user to control the quality of the classifier and providing greater resistance to adversarial attacks. The developed method is tested in comparison with conventional multiclass classifiers on the publicly available multidimensional data sets "Fisher's Irises" [13] and "Wheat Seeds" [14]. The ability to quantify the susceptibility to adversarial attacks using the EDCAP criteria is demonstrated.

Our contribution consists of two parts:

1    The proposal of a new multidimensional quality criterion EDCAP for multiclass classifiers.
2    The proposal of a new synthesis method for the multiclass classifier EnAE based on an ensemble of single-class classifiers.

## 2. Background

### 2.1. Review

The main criterion for the success of machine learning for image classification is to minimize the error on a given dataset. However, in the breakthrough work [15], it was revealed that deep neural networks have features of incorrect classification in the vicinity of the points of the training set, which can give a paradoxical result. At the same time, the term "adversarial example" was introduced for such input data, the points of which differ little from the points of the training sample but give an unexpected classification result.

In the subsequent work [1], the nature of adversarial examples in relation to deep neural networks was investigated, and simple methods for generating adversarial examples were proposed. The authors of the work note that the fact that classifiers based on neural networks have adversarial examples is largely due to their linear properties.

After the mentioned works, many researchers began to deal with the problem of finding adversarial examples and developing methods for protecting against adversarial attacks.

Currently, numerous tools have been developed that solve the problem of generating adversarial examples. At the same time, some of them operate in the mode of a well-known classification model (white-box). These methods include the fast gradient sign method (FGSM) [16], the Jacobian-based saliency map for finding the pixel that most influences the classification result [17], and the Carlini–Wagner attack [18]. If the classification model is unknown (black-box), then methods such as substitute attack [19], one-pixel attack [20], GAN-based method [21] are used to generate adversarial attacks. Note that the attacks cited above are strictly for images, but the general principles can be applied to any model. There are also such tools for implementing adversarial attacks as ART-IBM, FoolBox, CleverHans, ALFASVMlib, AdversariaLib, and MLsploit.

To protect against adversarial attacks on deep neural networks, various approaches have been proposed, such as replacing the ReLU activation function with Bounded-RELU [22], adversarial training [23], defensive distillation [24], and feature squeezing [25]. All these methods have common disadvantages, in that they do not protect against all types of attacks, slow down learning, and degrade the quality of learning. In addition, auxiliary neural networks used in some methods also represent a black box. Many experimental studies have been published on modern means of protection against adversarial examples [26], which reduce the probability of attack success [27,28]. There are also works that study the features of the data and, on this basis, build a criterion for identifying adversarial examples [29].

*2.2. Analysis*

An analysis of the literature shows that at the moment there is no generally accepted and effective solution to combat all types of adversarial attacks. Some even believe [30] that with high data dimensionality, the vulnerability of multilayer neural network models to adversarial attacks is inevitable.

The effectiveness of countermeasures against adversarial attacks is often measured in terms of the probability of detecting adversarial examples:

- "The study shows that defensive distillation can reduce the effectiveness of sample creation from 95% to less than 0.5% on a studied DNN" [24],
- "The results showed that our defending method achieved up to 96.3% classification accuracy on black-box adversarial examples, and detected up to 98.7% of the high confidence adversarial examples" [31],
- "The proposed method achieves 65% to 100% accuracy detecting adversarials with a wide range of attacks" [29].

In this case, the probability assessment is carried out for some implemented methods for searching for adversarial examples for the classifier of a particular data set. It turns out that the numerical assessment of the vulnerability of a classifier to adversarial attacks depends on the type of classifier and the method used to generate adversarial examples. This does not allow us to consider the calculated numerical score as a quality of the classifier itself, separated from the attack method.

At the same time, the use of classical classification quality scores calculated on the basis of the error matrix, such as accuracy, precision, recall, F-score, etc., cannot reveal the properties of the classifier outside the data set used for training and testing.

An interesting feature of adversarial attacks is their transferability between models that implement the attacked classifier [32]. At the same time, the models that implement the classifier can have a different structure, for example, one model is implemented by a deep neural network, and the other is implemented using SVM. This, as well as the ability to build adversarial examples without access to the attacked classifier (black-box), clearly hints that the reason for the existence of adversarial attacks lies in the data on which the target classifier is trained.

Even in [1], it was noted that the presence of adversarial attacks is largely due to the linear properties of classifiers in completely nonlinear problems solved by modern machine learning methods. Most researchers are currently passionate about studying deep neural networks. Therefore, the problem of adversarial attacks is mainly associated with deep neural networks, which are very far removed from linear models. At the same time, it seems obvious that the problem of having adversarial examples can be easily shown on simpler neural networks such as a multilayer perceptron.

Let us give an example of the result of scanning the feature space in a large neighborhood around the training data area of a neural network auto-encoder that implements the idea of a one-class classifier (Figure 1). Obviously, a large number of points that are far from the training set will be misclassified. In addition to the positive classification area around the training data, the neural network erroneously created another area away from the main one. Any point in this area will also be considered to belong to the target class.

This example emphasizes that adversarial examples can appear in simple neural networks and be a consequence of their non-linear properties.

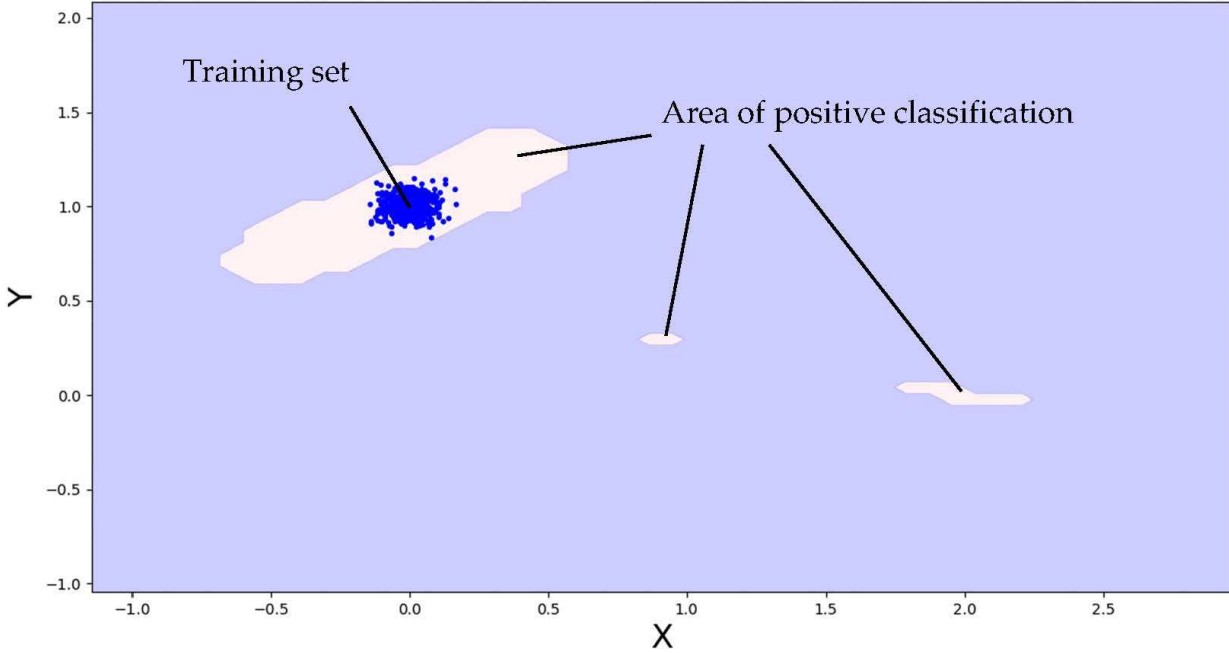

**Figure 1.** Areas of classification of the neural network auto-encoder.

It is important to systematically investigate the emergence of adversarial examples for classifiers of various natures, such as multilayer neural networks and SVMs. In this case it is necessary, if possible, to avoid the need to apply complex algorithms for generating adversarial examples. This will allow one to evaluate the quality of the classifier without reference to a specific method for finding adversarial examples. At the same time, it is not necessary to conduct research on multidimensional data sets, which include image databases such as MNIST, CIFAR, etc.

The synthesis of classifiers that are resistant to adversarial attacks is possible only with a given numerical quality criterion. The development of such a criterion should simplify the task of constructing adversarial-resistant classifiers and comparing different classifiers with each other.

Most machine learning methods for constructing classifiers have natural disadvantages resulting from the need to synthesize a computable function responsible for making decisions based on a data set that is unbalanced in terms of the number and density of points. At the same time, there are approaches that make it possible to make a decision based on an ensemble of classifiers [33], including one-class classifiers [34]. Usually, the voting approach is applied to classifiers of various models, as in [35], which makes it possible to obtain a more stable decision than is provided by each of the classifiers. For a number of reasons, it seems promising to use this approach to build a classifier with controlled resistance to adversarial attacks.

### 2.3. Motivation

We see that despite the large number of works devoted to adversarial attacks, a method for numerical assessment of the quality of classifiers from the standpoint of resistance to adversarial attacks has not been formulated. It seems important to develop and test an evaluation method that could be used to compare classifiers.

Also, as a result of the analysis of the literature, it was revealed that approaches to the synthesis of classifiers that are resistant to adversarial attacks are in most cases based on regularization for the purpose of more robust learning, which does not guarantee the desired property. In addition, resistance to adversarial attacks is tested empirically using

particular methods for generating adversarial examples, which cannot give an exhaustive picture of the classifier's behavior.

## 3. Methods

### 3.1. Designations

Further in the article, the notation given in Table 1 will be used.

**Table 1.** Designations.

| Symbol | Description |
|---|---|
| $X$ | A feature space as a set of objects to classify, possibly not countable |
| $\|X\|$ | The volume of continuous area |
| $X_T$, $X_T^0$ | A training subset of all objects |
| $X_T^{(i)}$ | A training subset of all objects of a class $i$ |
| $\hat{X}_T$ | A countable subset, available for training as usual in machine learning |
| $X_{Test}^{(i)}$ | A test subset of all objects of a class $i$ |
| $X_D$ | A set of objects which are positively classified by one-class classifier |
| $X_D^{(i)}$ | A set of objects that are classified as belonging to a class $i$ |
| $X_S$ | The regular $N$-dimension grid of points in the limited subset of $X$ |
| $X_T^*$ | All points of training set mapped to the closest grid points of $X_S$ |
| $X_D^*$ | A subset of grid points from $X_S$, which are positively classified |
| $Y$ | A set of class labels |
| $CL^1$ | A one-class classifier |
| $CL^K$ | A classifier of $K$ different classes which returns label id of a class |
| $CL^0$ | A common classifier which produces 1 if an object belongs any of defined classes and 0 otherwise |
| $CL^{(i)}$ | A classifier of class $i$ which produces 1 if an object belongs to class $i$ and 0 otherwise |
| $AE^0$ | An auto-encoder that determines membership to all labeled classes |
| $AE^{(i)}$ | An auto-encoder that determines membership to class $i$ |
| $NN_{n,h_1,...,h_k,m}$ | Neural network with $n$ inputs, $m$ outputs and $h_1, \ldots, h_k$ neurons in $k$ hidden layes |
| $RE$ | Reconstruction error: the discrepancy measure of an auto-encoder |

### 3.2. Basics of One-Class Classification Quality and Adversarial Samples

Consider the problem of synthesizing a classifier using the machine learning method. For simplicity, we take the problem of one-class classification. Denote by $X$ the set of classified objects. For a one-class classifier, the output is $Y = \{0, 1\}$, with 1 indicating membership in the target class and 0 indicating no membership. The subset corresponding to the target class will be denoted as $X_T \subset X$. Allow the object membership function to be set to the target class:

$$\forall x \in X : f(x) = \begin{cases} 1, & x \in X_T \\ 0, & x \notin X_T \end{cases}$$

This function is an ideal one-class classifier, however, in practice, as a rule, it is not possible to obtain all $x \in X_T$ values and use them for training. Usually, a limited countable training set $\hat{X}_T \subset X_T$ is available, from which a training sample is formed, as well as samples to control the training and testing process. Denote the trained one-class classifier:

$$CL^1(x) : X \to Y$$

Despite the fact that $CL^1$ is trained on $\hat{X}_T \subset X_T$, as a result of the synthesis, various situations are possible that correspond to the possible outcomes of the binary classification (Table 2): True Positive ($TP$), True Negative ($TN$), False Positive ($FP$), False Negative ($FN$).

**Table 2.** Binary Classification Outcomes.

| Classification\Class | Positive $x \in X_T$ | Negative $x \notin X_T$ |
|---|---|---|
| Positive | $TP: CL^1(x) = 1$ | $FP: CL^1(x) = 1$ |
| Negative | $FN: CL^1(x) = 0$ | $TN: CL^1(x) = 0$ |

In fact, the real one-class classifier differs from the ideal one in that the set of objects that it refers to the target class is deformed, that is, it differs from the target $X_D \neq X_T$:

$$\forall x \in X: \ CL^1(x) = \begin{cases} 1, & x \in X_D \\ 0, & x \notin X_D \end{cases}$$

A similar situation can be visualized in Figure 2.

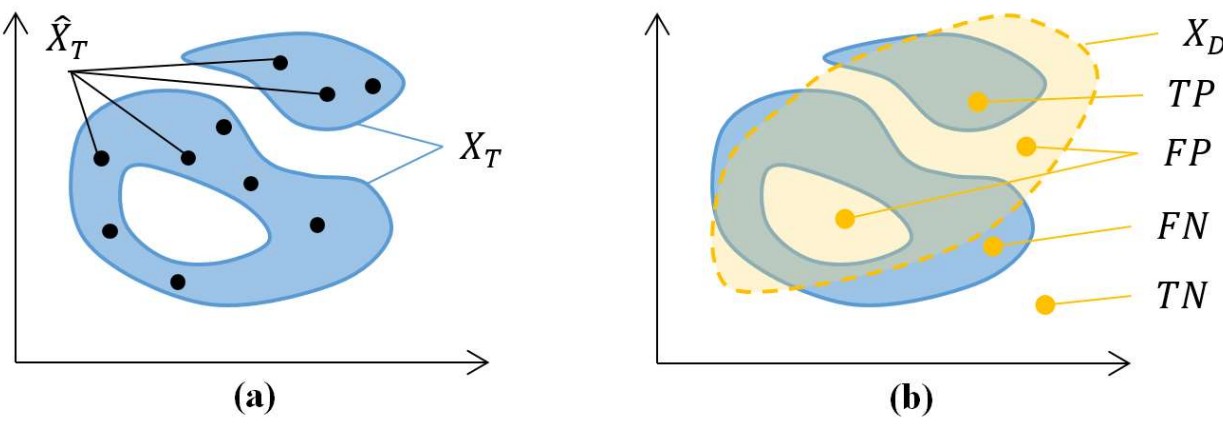

**Figure 2.** Target class area (**a**) and deformed area (**b**).

The quality of a one-class classifier is the closeness of $CL^1$ to the ideal $f$, which can be expressed in terms of the properties of the $X_D$ set compared to the $X_T$ set. In particular, the classification outcomes $FP$ and $FN$ differ from the ideal. Situation FP corresponds to the case $X_D \backslash X_T \neq \varnothing$, and situation FN corresponds to the case $X_T \backslash X_D \neq \varnothing$, where the operation\denotes the subtraction of sets. For continuous sets of classification points, we can introduce the concept of volume, for which we use the notation $|X|$, and $|\varnothing| = 0$. In this case, the presence of situations $FP$ and $FN$ can be described in terms of volume as follows:

$$FP: |X_D \backslash X_T| \geq 0$$
$$FN: 0 \leq |X_T \backslash X_D| \leq |X_T|$$

Note that the adversarial examples for a one-class classifier form the set $FP$. The outcome of a correct classification is described in terms of volume:

$$TP: \ 0 \leq |X_T \cap X_D| \leq |X_T|$$

Moreover, $|X_T \backslash X_D| = 0$ follows from $X_T \cap X_D = X_T$.

It is desirable that the numerical assessment of the quality of the classifier does not depend on the volume of the target set, which will make it possible to compare the quality of classifiers trained for various applied problems. Let us introduce quality indicators normalized to the volume of the target set:

$$Excess = \frac{|X_D \backslash X_T|}{|X_T|}$$

$$Deficit = \frac{|X_T \backslash X_D|}{|X_T|}$$

$$Coating = \frac{|X_T \cap X_D|}{|X_T|}$$

The quality indicator *Excess* takes on a value greater than 0 if $CL^1$ incorrectly classifies objects outside the target set. The quality indicator *Deficit* takes a value from 0 to 1, and if $CL^1$ incorrectly classifies objects within the target set, then the metric is greater than 0. The limit value $Deficit = 1$ is the situation when the target and classified sets do not intersect. The *Coating* quality indicator takes values from 0 to 1, and the value 0 corresponds to the zero intersection of the sets $X_T$ and $X_D$, which is difficult to imagine with any effective machine learning procedure, and 1 corresponds to the usual situation when the classifier makes the right decision on the entire area of training data.

The target set approximation accuracy can be estimated as the ratio of the volume of the target set to the volume deformed by the classifier:

$$Approx = \frac{|X_T|}{|X_D|}$$

An ideal one-class classifier is characterized by the following values of quality indicators:

$$Excess = 0, \quad Deficit = 0, \quad Coating = 1, \quad Approx = 1$$

Taken together, the proposed quality indicators form a multidimensional criterion, which for brevity we will denote as EDCA. This criterion was introduced by the authors in [36] in order to numerically evaluate the quality of an anomaly detector based on a neural network auto-encoder.

In reality, the set of points in the training dataset is discrete. In order to be able to calculate the proposed quality indicators for real classifiers, it is necessary to introduce a grid approximation of a continuous space with the restriction of a certain work area in which we will calculate the quality indicators. By limiting the space to a workspace that obviously includes sets of training and test data, we can normalize the coordinates in the range from 0 to 1. Let us introduce a uniform grid with step h over all coordinates of the limited grid space. Thus, for an *N*-dimensional feature space, we will have $\left(\frac{1}{h}\right)^N$ grid space cells. Let us call the set of these cells the scan set $X_S$. We will attribute the points of the sets $X_T$ and $X_D$ to the cells of the scan set to which they belong. Thus, the cells of the training data form the set $X_T^* \subset X_S$, and the set $X_D^* \subset X_S$ will be formed from cells with a positive classification result for the point in the center of the cell.

It is recommended to choose the cell size for the data set under study so that the number of training set points in each cell is balanced and tends to 1. If the cell size is too small, then in the case of high data dimension, the calculations will require significantly more time. In addition, a too small value of *h* will lead to the fact that neighboring $X_T$ points will be separated in the scan set by several empty cells that do not belong to $X_T^*$, which can lead to the loss of significance of the EDCA numerical estimates.

An example of EDCA numerical estimates for a single-class classifier is shown in Figure 3.

Values *Deficit*, *Coating* largely characterize the quality of classification in the classical sense.

The *Excess* characteristic is the most important for assessing the vulnerability of a classifier to adversarial attacks. This characteristic shows the size of the area of positive classification in the feature space outside the boundaries of the area of training examples. Figure 3 shows that the larger this area, the larger the value of *Excess*. Comparing the *Excess* characteristics of different classifiers allows one to compare the risk of adversarial attacks, quantified by the amount of feature space in which examples that cause classifier errors can be found. It should be noted that the calculation of *Excess* is performed without knowledge of the classifier mechanisms (black-box) and can be used to compare classifiers of different nature.

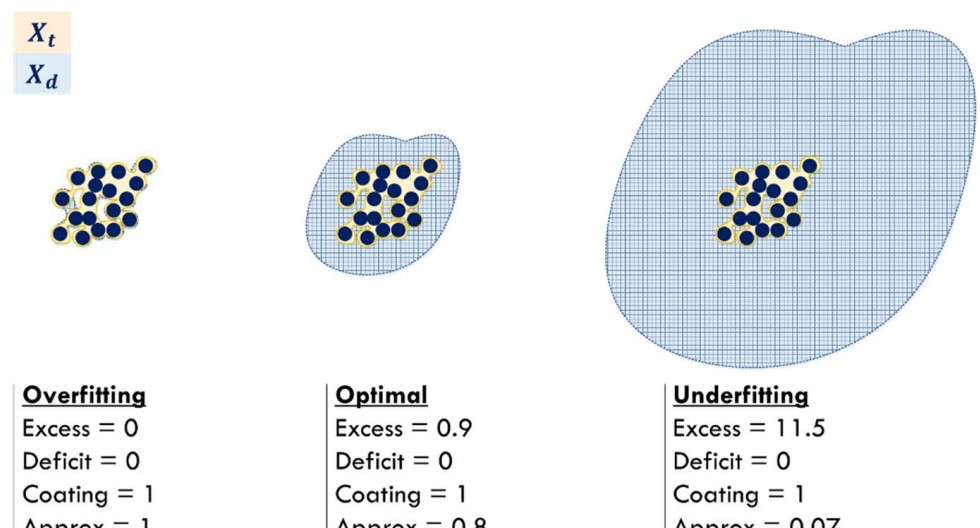

**Figure 3.** The relationship between the size of the $X_D^*$ area and the values of the EDCA characteristics.

The *Approx* characteristic shows how closely the area of positive classification matches the training set. In case the classifier is too promiscuous and prone to adversarial attacks, the *Approx* value is close to 0.

It should be noted that the "ideal" classifier, characterized by the limiting values of the characteristics, is vulnerable to the FN classification errors in the case of a non-representative training sample. Classification errors like this can also be considered adversarial attacks in some way.

If, due to the specifics of the problem being solved, FP errors are more critical than errors of the second kind, then it is recommended to choose a classifier that provides the minimum *Excess* value.

### 3.3. Principles of the Proposed Criteria for Assessing Quality of Multiclass Classifiers

As noted earlier, classical classification quality estimates based on an error matrix cannot fully characterize the properties of a classifier synthesized by machine learning methods. To solve the problem of assessing the resistance of a classifier to adversarial attacks, it is proposed to analyze the behavior of the trained classifier in the area of the feature space in the vicinity of the training set. Consider the estimation method on the example of a classifier trained on the training data set $X_T$. Let us say the training set includes $M$ classes. Let us denote the set of examples from the training set corresponding to class $i$ as $X_T^{(i)}$, where $i = \overline{1, M}$. The set of samples that the trained classifier recognizes as class $i$ will be denoted as $X_D^{(i)}$.

For a numerical evaluation of the properties of a multiclass classifier, a multidimensional EDCA criterion can be used, which includes the previously introduced characteristics for each of the classes. In this case, for each of the classes, their own characteristics $Excess^{(i)}$, $Deficit^{(i)}$, $Coating^{(i)}$, $Approx^{(i)}$ can be calculated. This criterion was first used to assess the quality of a classifier developed for dynamic classification under concept drift [37] and was also studied for comparison with classical quality estimates based on an error matrix [38].

For the case of multiclass classification, a characteristic feature is also the partial overlap of areas containing examples belonging to different classes. For example, the Iris data set [13] has such a feature. In this case, the construction of an ideal classifier seems impossible in principle, which is well known to machine learning researchers.

A characteristic of the ambiguity of the training data set is a non-zero intersection of the regions of classes $i$ and $j$:

$$X_T^{(i)} \cap X_T^{(j)} \neq \varnothing$$

In the domain of ambiguous classifier definition, any example will be adversarial in some sense. A reasonable characteristic for evaluating the behavior of a multiclass classifier in the area of ambiguity can be an assessment of its preference $Pref^{(i)}$ for one class over all the others. For the intersection of two classes, such a characteristic can be calculated as the ratio of the volume of the domain of belonging to a particular class to the volume of the domain of ambiguous definition:

$$Pref^{(1)} = \begin{cases} 0, & X_T^{(1)} \cap X_T^{(2)} = \varnothing \\ \dfrac{\left|\left(X_T^{(1)} \cap X_T^{(2)}\right) \cap X_D^{(1)}\right|}{\left|X_T^{(1)} \cap X_T^{(2)}\right|}, & X_T^{(1)} \cap X_T^{(2)} \neq \varnothing \end{cases}$$

Graphically, the preference area is shown in Figure 4.

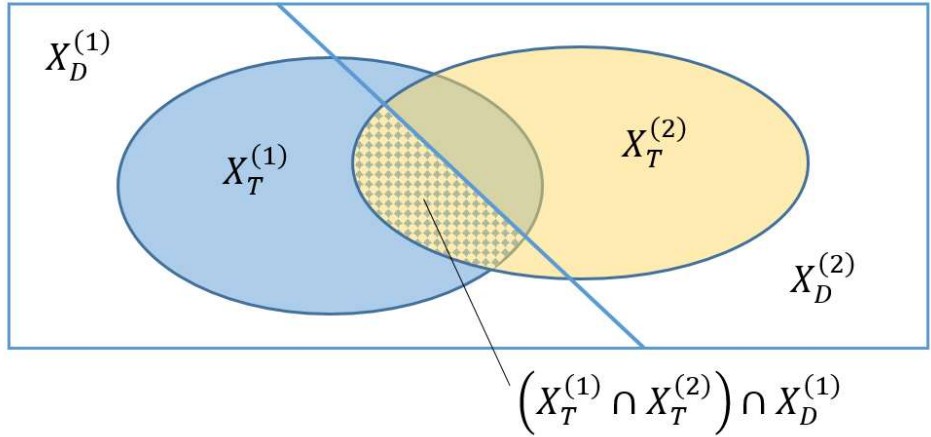

**Figure 4.** Example of a Class 1 preference area in the ambiguity region $\left(X_T^{(1)} \cap X_T^{(2)}\right) \cap X_D^{(1)}$.

Generalizing the definition for the case of intersections of the domains of definition of several classes, we get:

$$Pref^{(i)} = \begin{cases} 0, & \cup_{\forall j \neq i} X_T^{(i)} \cap X_T^{(j)} = \varnothing \\ \dfrac{\left|\cup_{\forall j \neq i}\left(X_T^{(i)} \cap X_T^{(j)}\right) \cap X_D^{(i)}\right|}{\left|\cup_{\forall j \neq i} X_T^{(i)} \cap X_T^{(j)}\right|}, & \cup_{\forall j \neq i} X_T^{(i)} \cap X_T^{(j)} \neq \varnothing \end{cases}$$

The value of $Pref^{(i)}$ changes from 0 to 1 and shows how the classifier in areas of ambiguity tends to make a decision in favor of class $i$. For practical application, this characteristic should be calculated for the cells of the $X_S$ scan set.

Thus, to numerically characterize the properties of a multiclass classifier, it is proposed to calculate for each of the classes $Excess^{(i)}$, $Deficit^{(i)}$, $Coating^{(i)}$, $Approx^{(i)}$, $Pref^{(i)}$.

The use of a quality criterion, the size of which depends on the number of classes, seems inconvenient. At the same time, if we do not have preferences for the significance of a particular class, then a tuple of 5 values of $Excess$, $Deficit$, $Coating$, $Approx$, $Pref$ can be considered a reasonable numerical assessment of the quality of the classifier, each of which characterizes the classifier in the worst way for all classes:

$$Excess = \sum_i Excess^{(i)}$$

$$Deficit = \max_i Deficit^{(i)}$$

$$Coating = \min_i Coating^{(i)}$$

$$Approx = \min_i Approx^{(i)}$$

$$Pref = \max_i Pref^{(i)}$$

For the characteristics *Deficit*, *Coating*, *Approx* and *Pref*, the worst value (minimum or maximum) is taken, and for *Excess*—the sum of all classes. The set of proposed characteristics form a multidimensional EDCAP quality criterion, which can be used to compare classifiers, taking into account the possible intersection of the definition areas of some classes. The calculation of EDCAP characteristics must be carried out on the same $X_S$ scan set, that is, with the same working area and grid spacing.

Let an adversarial attack on class *i* be an example that is different from the examples belonging to class *i*, but classified as belonging to it. Then, to assess the vulnerability of the classifier to adversarial attacks on class *i*, the most valuable characteristic is $Excess^{(i)}$, however, in cases of overlapping areas, it is also necessary to analyze $Pref^{(i)}$.

In this case, the task of comparing two classifiers in terms of their vulnerability to adversarial attacks can be reduced to comparing the calculated *Excess* values in the case of non-overlapping training sets. A classifier with a lower *Excess* value has a smaller area containing adversarial samples for all classes.

### 3.4. Principles of the Proposed Method of Synthesis for Adversarial Attack-Resistant Classifiers

Consider the problem of designing a classifier resistant to adversarial attacks. To assess stability, we will use the EDCAP criterion. First, it is necessary to analyze the properties of traditional classifiers obtained as a result of machine learning. To do this, it is convenient to use the approach used to calculate the EDCAP quality indicators with visualization of the $X_T^*$ and $X_D^*$ areas for all classes. Typical cases are shown in Figure 5.

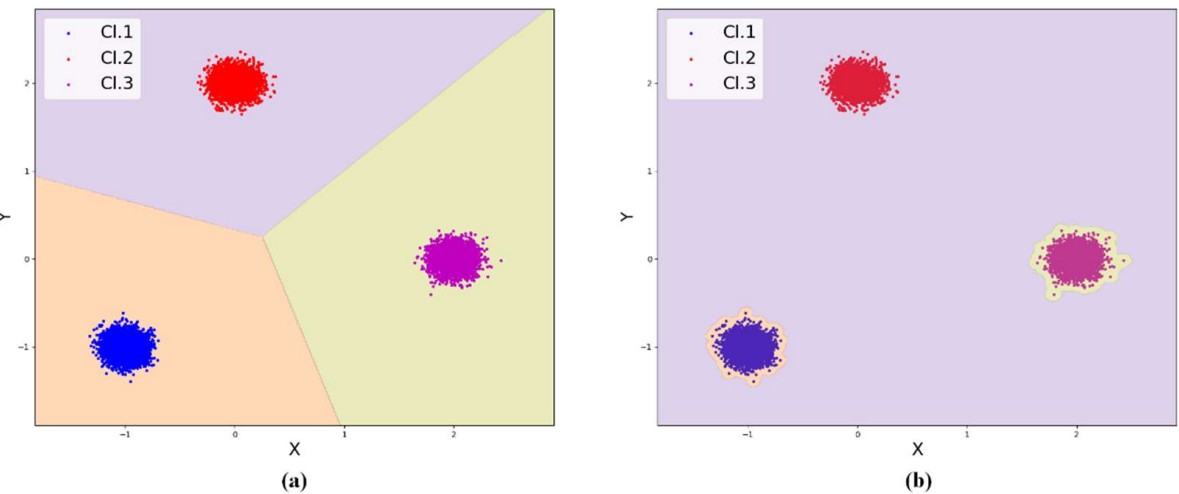

**Figure 5.** The first (**a**) and second (**b**) feature space separation variant of machine learning classifiers.

Such options for splitting the feature space after training are the cause of classifier errors. For example, in case Figure 5a any example, even far enough away from the scope of the class given by the training set, will be classified as belonging to this class. The use of such a classifier is undesirable for a number of practical tasks, for example, in intrusion detection systems. In case Figure 5b, examples close to the scope of classes Cl.1 and Cl.3 will be classified as belonging to class Cl.2. On the classifier in case Figure 5b, it is possible to generate a set of adversarial examples that cause false positive errors for classes Cl.1 and Cl.3, and a set of adversarial examples that cause false negative errors for class Cl.2. The use of such a classifier in computer vision and autopilot systems can lead to serious consequences due to the wrong decision.

The described situations can be avoided if an ensemble of auto-encoders is used for classification. Each ensemble auto-encoder in the learning process seeks to build a compact region that spans many training examples of the target class. Let us construct a multiclass classification algorithm based on an ensemble of auto-encoders. Each of the auto-encoders

should be trained only on examples of one of the classes. Examples from other classes for this auto-encoder are anomalies. The main property of the auto-encoder is a small reconstruction error for examples from the training set and those close to them. A large reconstruction error indicates that similar examples did not appear in the learning process. To separate familiar examples from unfamiliar examples, the reconstruction error threshold $RE_{th}$ is used, which determines the sensitivity of the algorithm to novelty.

In addition to class-specific auto-encoders, it is useful to introduce an auto-encoder $AE^0$ that will separate examples of known classes from examples that are unlike any of the classes. Such an auto-encoder acts as an anomaly detector and is an additional defense against adversarial attacks.

An example of such division of the feature space using an ensemble of auto-encoders based on neural networks is shown in Figure 6.

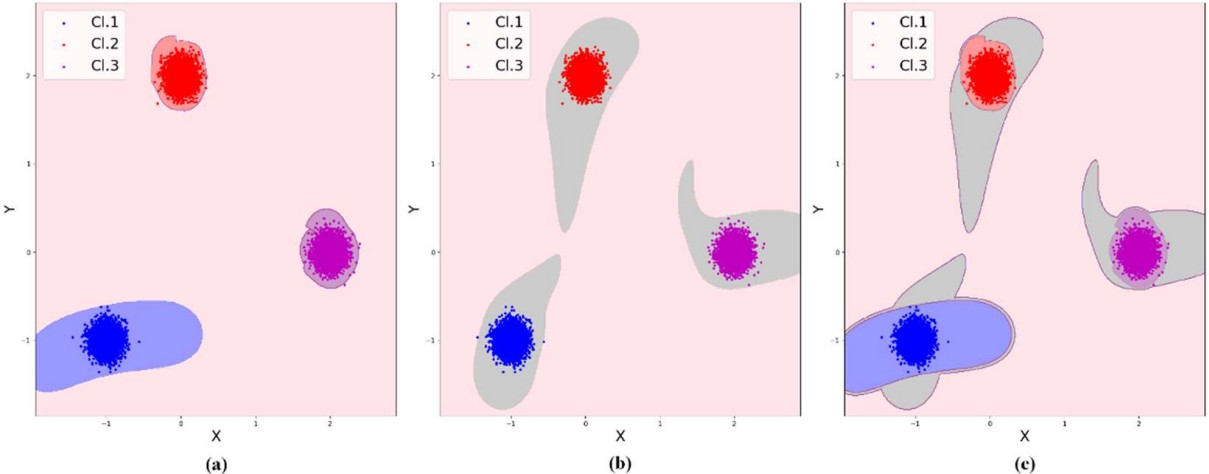

**Figure 6.** Feature space separation after training an ensemble of auto-encoders: specific class auto-encoders $AE^{(i)}$ (**a**), all classes auto-encoder $AE^0$ (**b**), combined effect of all auto-encoders (**c**).

Let us calculate the EDCAP characteristics for the considered example and three variants of the classifier, including an ensemble with a common auto-encoder. Since the data set certainly does not contain intersections of the class definition area, the *Pref* characteristic will be equal to zero.

We present the values of the characteristics *Excess* and *Approx* of individual classes in Table 3.

**Table 3.** Excess and Approx values for individual classes.

| Variant of the Classifier | Excess | | | Approx | | |
|---|---|---|---|---|---|---|
| | **Cl.1** | **Cl.2** | **Cl.3** | **Cl.1** | **Cl.2** | **Cl.3** |
| Figure 5a | 16.09 | 20.49 | 20.86 | 0.06 | 0.05 | 0.04 |
| Figure 5b | 0.36 | 54.13 | 0.40 | 0.74 | 0.02 | 0.71 |
| Figure 6 | 3.51 | 0.43 | 0.53 | 0.22 | 0.71 | 0.66 |
| Figure 6 (for all classes) | 2.72 | | | 0.27 | | |

A generalized characteristic of the considered classifiers according to the EDCA criterion is presented in Table 4.

It is easy to see from the tables that traditional classifiers have the highest value of the Excess characteristic, which means a higher vulnerability to adversarial attacks. A similar conclusion was made earlier based on visual analysis of Figure 5.

Another important conclusion is the increased resistance of the ensemble of auto-encoders to adversarial attacks, which can be improved with the addition of a common auto-encoder trained on the data of all classes.

**Table 4.** EDCA Quality Characteristics for the Considered Classifiers.

| Variant of the Classifier | Excess | Deficit | Coating | Approx |
|---|---|---|---|---|
| Figure 5a | 57.44 | 0 | 1 | 0.04 |
| Figure 5b | 54.53 | 0 | 1 | 0.02 |
| Figure 6 | 4.86 | 0 | 1 | 0.24 |
| Figure 6 (for all classes) | 1.75 | 0 | 1 | 0.36 |

*3.5. The Synthesis and the Algorithm of an Adversarial Attack-Resistant Classifier*

Let us consider the structure, functioning algorithm and synthesis method of the EnAE (Ensemble of Auto-Encoders) multiclass classifier based on an ensemble of neural network auto-encoders. This classifier also includes a general auto-encoder for the union of all classes, which allows the classifier to detect the absence of belonging to known classes. Applying control of the classification area outside the training set based on the EDCAP approach is supposed to make this classifier more resistant to adversarial attacks.

Let us introduce the necessary notation and definitions.

To designate a neural network of the multilayer perceptron type with $n$ inputs, hidden fully connected layers with the number of neurons $h_1, \ldots, h_k$ and the number of outputs $m$, we will use the notation:

$$y = NN_{n,h_1,\ldots,h_k,m}(x), \quad x \in R^n, \, y \in R^m$$

To designate the Dropout layer, we will indicate the letter $D$ in the notation:

$$y = NN_{n,h_1,D,h_2,D,m}(x), \quad x \in R^n, \, y \in R^m$$

We will call an auto-encoder a neural network of the multilayer perceptron type with an equal number of inputs and outputs $n$ and more than one hidden layer:

$$\widetilde{x} = AE(x) \equiv NN_{n,\ldots,n}(x), \quad x, \widetilde{x} \in R^n$$

The result of the auto-encoder is a reconstruction error (*RE*):

$$RE(x) = \sqrt{\sum_{j=1}^n \left(x_j - \widetilde{x}_j\right)^2}, \quad \widetilde{x} = AE(x)$$

For examples of the training set and similar sets, the reconstruction error should be small, and for other examples it should be large. The training of the auto-encoder on the training set $X_T$ is carried out according to the criterion of minimizing the reconstruction error:

$$Training\{AE, \, X_T\} : \sum_{x \in X_T} RE(x) \rightarrow min$$

Setting the reconstruction error threshold $RE_{th}$ allows one to build a decision rule for a one-class classifier:

$$CL^1(x) = \begin{cases} 0, & RE(x) > RE_{th} \\ 1, & RE(x) \leq RE_{th} \end{cases}$$

A possible way to calculate the threshold is the maximum reconstruction error on the training set:

$$RE_{th} = \max_{x \in X_T} RE(x)$$

Let $M$ classes given in the vector space $X \subseteq R^n$ and described by training sets of samples:

$$X_T^{(i)} \subset X, \quad i = \overline{1, M}$$

For convenience, let us introduce the training set of samples of all classes:

$$X_T^0 = \cup_{i=1}^M X_T^{(i)}$$

Suppose there are also test sets for each of the classes:

$$X_{Test}^{(i)} \subset X, \quad i = \overline{1, M}$$

We will assume that the data $X_T^{(i)}$ and $X_{Test}^{(i)}$ have passed the necessary preprocessing and normalization.

We need to build a classifier $EnAE(x)$ that would report the class label $i$ for any $x \in X$ if $x$ is similar to the examples from $X_T^{(i)}$ or 0 if $x$ is not similar to any of the classes.

### 3.5.1. The Synthesis of the EnAE Classifier

The synthesis procedure for the $EnAE(x)$ classifier is presented below:

1    Setting architecture $AE^0, AE^{(i)}, i = \overline{1, M}$
2    Training of neural network auto-encoders:

    a    $Training\{AE^0, X_T^0\}$
    b    $Training\{AE^{(i)}, X_T^{(i)}\}, i = \overline{1, M}$

3    Calculation of the values that determine decision-making:

    a    Threshold calculation $RE_{th}^0 = \max\limits_{x \in X_T^0} RE^0(x)$ for single-class classifier $CL^0(x)$ based on $AE^0$

    b    Threshold calculation $RE_{th}^{(i)} = \max\limits_{x \in X_T^{(i)}} RE^{(i)}(x)$ for single-class classifier $CL^0(x)$ based on $AE^{(i)}, i = \overline{1, M}$

4    Calculation of the quality characteristics of the resulting classifier with details by class:

    a    According to the EDCAP criterion: $Excess^{(i)}, Deficit^{(i)}, Coating^{(i)}, Approx^{(i)}, Pref^{(i)}, i = \overline{1, M}$
    b    By test sets $X_{Test}^{(i)}$: $Precision^{(i)}, Recall^{(i)}, Fscore^{(i)}, i = \overline{1, M}$

5    Analysis of the classification quality characteristics of individual auto-encoders $AE^0, AE^{(i)}, i = \overline{1, M}$, adjusting their architecture, training parameters and repeating steps 2–4 until the required quality level is obtained.

### 3.5.2. The Algorithm of the EnAE Classifier

The $EnAE(x)$ auto-encoder operation algorithm is presented below:

1    If $CL^0(x) = 0$, then $x$ does not belong to any of the classes.
2    Otherwise, if $CL^{(i)}(x) = 0 \; \forall i : 1 \le i \le M$, then $x$ does not belong to any of the classes.
3    Otherwise, if there is only one $\exists! i : 1 \le i \le M$ such that $CL^{(i)}(x) = 1$, then $x$ belongs to class $i$.
4    Otherwise, there are several $i = k_1, k_2, \ldots$ such that $CL^{(i)}(x) = 1$, then the example $x$ belongs to the class $k_j$ for which the relative reconstruction error is minimal:

$$k_j = \arg\min\limits_{i=k_1,k_2,\ldots} RE^{(i)} / RE_{th}^{(i)}$$

The proposed synthesis method uses a neural network auto-encoder, however, it is worth noting that a one-class classifier can be implemented based on another machine learning method, for example, SVM. The chosen machine learning method should build a compact region for each class in the feature space.

### 4. Case Study

This section presents the results of studying the resistance of classifiers to adversarial attacks. Consider a traditional classifier based on SVM, an ensemble of classifiers implemented as a soft voting classifier (VC) [39] and an EnAE classifier synthesized according to the method described above. The SVM and VC classifiers were trained using the sklearn library, and the EnAE classifier was trained using the keras library. The experiments were carried out on well-known datasets.

To assess the quality, both the traditional characteristics precision, recall, F-score, and the EDCAP proposed above were used. The actual resistance to adversarial attacks was tested based on adversarial examples created using the procedure described below.

#### 4.1. The Method of Adversarial Samples Generation

According to the generally accepted definition, an adversarial example is a minimally modified example of a known class for which the machine learning model produces an incorrect prediction.

To find such examples, it is proposed to generate a set of minimally modified examples of the training data set. Examples of such a set, classified incorrectly, are adversarial examples for this classifier.

In this study, for each example of the training set, 10 modified examples were generated with feature dimension $N$. The modification of the example of the training set $x \in X_T$ consisted in replacing one of the features $x_j$ with one of 10 uniformly distributed values in the range from $x_j - 0.05$ to $x_j + 0.05$. Thus, the created test example differed from the original one in only one feature. If the trial example led to a different classification result than the original one, this example was recognized as adversarial.

Here is the initial example $x$ from the training sample of class "3" of the Wine dataset [40] and the adversarial example $x^*$ obtained by modifying the first feature:

$$x = \{0.7500;\ 0.8498;\ 0.4652;\ 0.4845;\ 0.1087; \ldots\}$$
$$x^* = \{0.7889;\ 0.8498;\ 0.4652;\ 0.4845;\ 0.1087; \ldots\}$$

The synthesized classifier makes a mistake and classifies the example $x^*$ as class "2". Such adversarial examples are reminiscent of attacks on image classifiers called one-pixel attacks [20]. It is important to note that there are usually thousands of such examples. And as the dimension of the feature space grows, the number of potential adversarial examples increases.

This method of generating adversarial examples was used in experiments for two well-known data sets: Iris [13] and Seeds [14].

#### 4.2. Synthesis of Classifiers on the Data Set Iris and Evaluation of Their Resistance to Adversarial Attacks

Let us synthesize and compare the SVM, the VC and EnAE classifiers for the Fisher's Irises data set [13]. As is known, this set has a four-dimensional feature space and three classes. Let us divide the initial data set into training (80%) and test (20%) sets.

For clarity, the training data set is represented in two-dimensional space using the multidimensional data visualization algorithm—T-SNE (Figure 7).

The synthesis of the SVM classifier was carried out with the parameters $gamma = 100$, $C = 10$. The VC is implemented based on a linear support vector machine, a decision tree, and a K-nearest neighbor classifier. In the EnAE classifier, all auto-encoders had an architecture:

$$NN_{4,4,3,2,3,4,4}$$

The ADAM learning algorithm is used for training all auto-encoders. The learning rate is set to $\eta = 0.001$. Each auto-encoder was trained for $3 * 10^5$ epochs. The calculated traditional quality characteristics on the test sample are presented in Table 5.

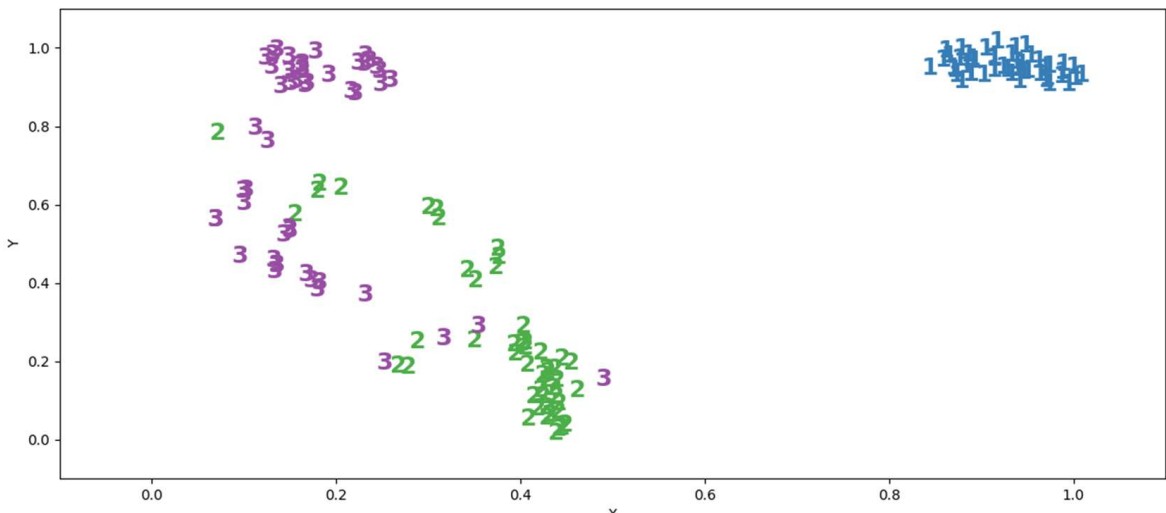

**Figure 7.** Fisher's Irises training dataset (different classes colored differently for convenience).

**Table 5.** SVM, VC and EnAE quality assessment by traditional criteria.

| Classifier | Precision | Recall | F-Score |
|:---:|:---:|:---:|:---:|
| SVM | 0.97 | 0.97 | 0.97 |
| VC | 0.97 | 0.97 | 0.97 |
| EnAE | 1.00 | 0.97 | 0.98 |

To calculate the EDCAP characteristics, the scan set was determined with a step $h = 0.14$. The calculated values of the EDCAP characteristics for both classifiers are in Table 6.

**Table 6.** SVM, VC and EnAE quality assessment by EDCAP.

| Classifier | Excess | Deficit | Coating | Approx | Pref |
|:---:|:---:|:---:|:---:|:---:|:---:|
| SVM | 93.94 | 0.00 | 1.00 | 0.01 | 1.00 |
| VC | 178.9 | 0.33 | 0.66 | 0.01 | 1.00 |
| EnAE | 16.82 | 0.00 | 1.00 | 0.08 | 1.00 |

An analysis of the values of the EDCAP characteristics for each class in the case of the SVM classifier showed that the entire area outside the compact areas of classes "1" and "2" belongs to class "3". This space partitioning option is of the type shown in Figure 2b. The values of the EDCAP characteristics for each class showed that the space partitioning option in the case of a VC classifier is closer to the type shown in Figure 2a.

$Pref = 1$ means that some classes have an ambiguous definition by training sets. In that case an exactly one cell of scan set is shared between training sets for two different classes. The ambiguity is resolved by classifiers to the favor of class "2" for SVM, VC and to the class "3" for EnAE.

It can be noted that, despite the high quality of classification by traditional criteria, the characteristics of EDCAP show that the SVM classifier, like the VC classifier, should be more vulnerable to adversarial examples. And the $Pref$ feature reveals that the SVM classifier is more vulnerable to adversarial attacks on the class on "3".

To check the actual stability of the classifiers to the adversarial attacks, 3280 test examples were generated according to the algorithm described in Section 4.1. As a result of the classification of trial examples, the number of successful adversarial attacks for each classifier and each class was counted. The results of the experiment are shown in Table 7.

**Table 7.** Number of successful adversarial attacks against SVM, VC and EnAE.

| Classifier | Number of Adversarial Attacks | | |
| --- | --- | --- | --- |
| | Class "1" | Class "2" | Class "3" |
| SVM | 0 | 0 | 115 |
| VC | 0 | 0 | 97 |
| EnAE | 0 | 0 | 18 |

For clarity, Figure 8 shows the training sample and successful adversarial attacks on the SVM classifier in the compressed feature space.

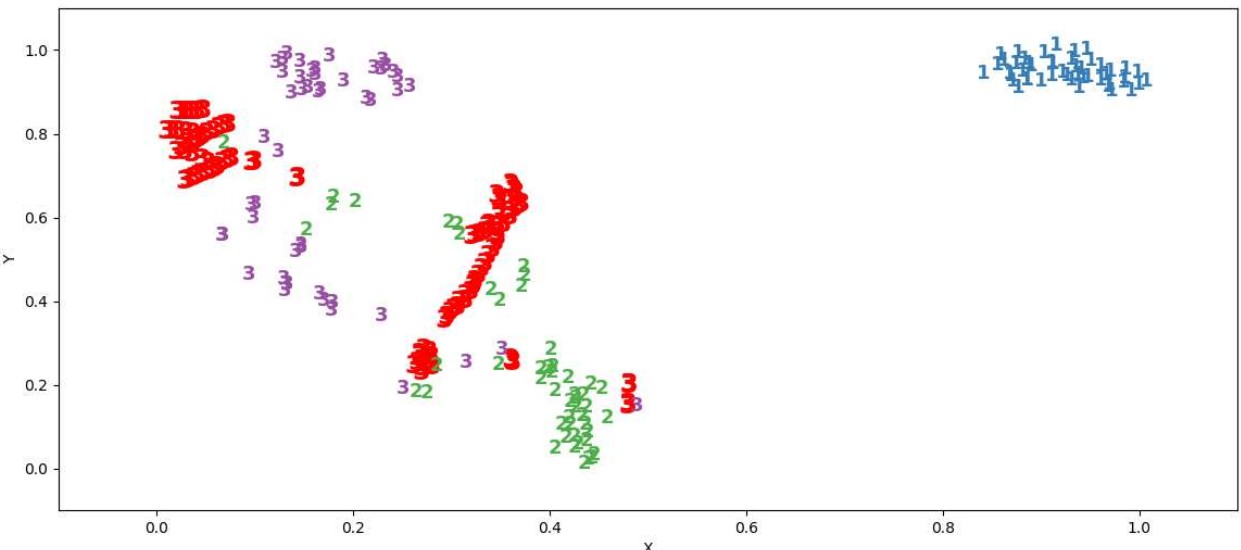

**Figure 8.** Successful adversarial attacks on the SVM classifier (highlighted in red).

Figure 9 shows successful adversarial attacks on the VC classifier.

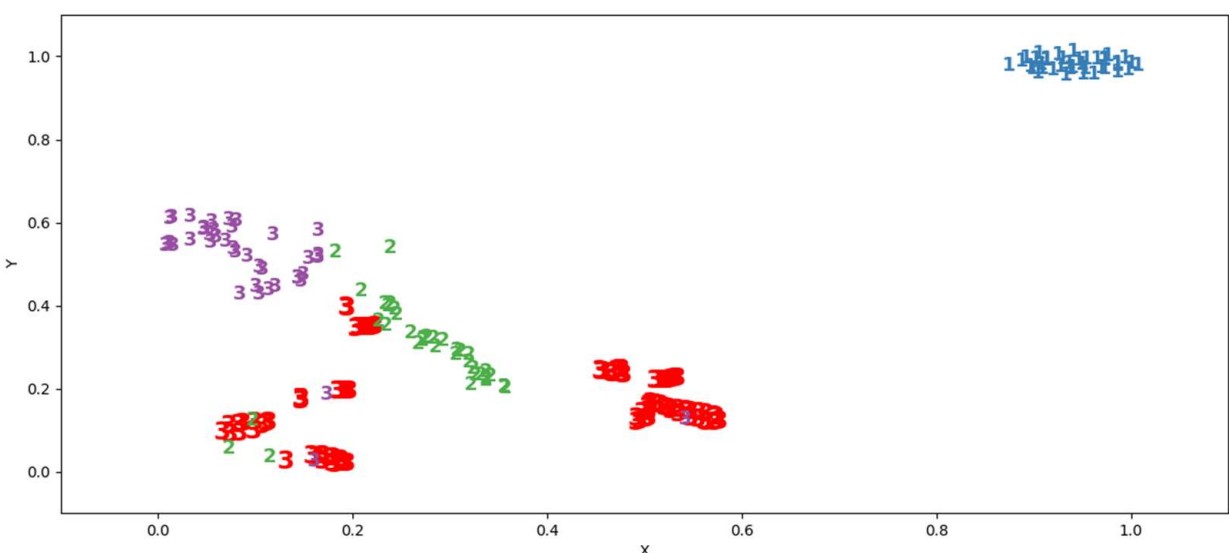

**Figure 9.** Successful adversarial attacks on the VC classifier (highlighted in red).

Figure 10 shows the training set and successful adversarial attacks on the EnAE classifier.

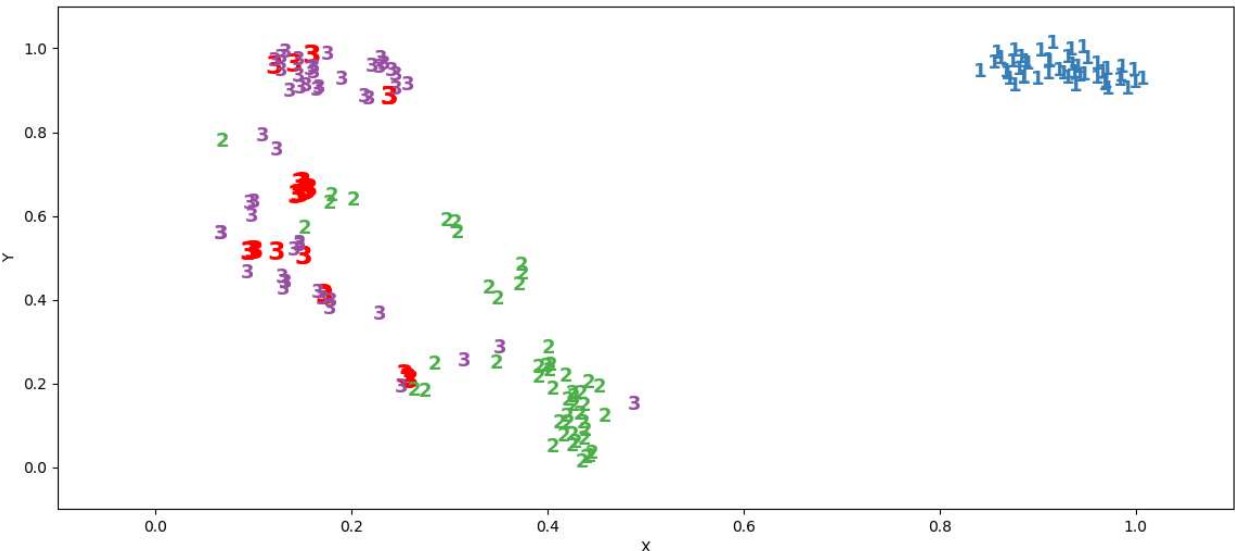

**Figure 10.** Successful adversarial attacks on the EnAE classifier (highlighted in red).

The obtained results of the actual resistance of the studied classifiers to adversarial attacks were evaluated using traditional classification quality metrics. The scores were calculated on a set of test examples marked with class labels from the training data set. A successful result of the classification was considered to be the receipt of the same label for the test case. If the result of the classification differed from the expected one, then such an example was considered adversarial and demonstrated the vulnerability of the classifier. The evaluation results for the SVM, the VC and EnAE classifiers are shown in Table 8.

**Table 8.** Results of assessing the quality of the classification of adversarial examples.

| Classifier | Precision | Recall | F-Score |
|---|---|---|---|
| SVM | 0.95 | 1.00 | 0.95 |
| VC | 1.00 | 0.97 | 0.98 |
| EnAE | 0.98 | 1.00 | 0.99 |

The obtained results of the experiment confirm that the classifier synthesized by the proposed method turned out to be more resistant to adversarial examples.

### 4.3. Synthesis of Classifiers on the Data Set Seeds and Evaluation of Their Resistance to Adversarial Attacks

Let us take another data set Seeds [14] and repeat the experiments on the synthesis and comparison of two types of classifiers: a deep neural network (DNN) and an ensemble of auto-encoders EnAE. This dataset has three classes defined by feature vectors with dimension 7. Figure 11 shows a 2D visualization of this dataset using the T-SNE algorithm.

For classification in this dataset, a neural network DNN of the multilayer perceptron type was created, including four fully connected layers, Dropout layers and a Softmax output layer of three neurons. The DNN network architecture is shown below:

$$y = NN_{7,7,D,5,D,3,D,3}(x), \ x \in R^7, \ y \in R^3$$

DNN training was carried out for $5 * 10^4$ epochs.

Also for this data set, the EnAE classifier was synthesized, which includes four neural network auto-encoders. The architecture of each auto-encoder is listed below:

$$y = NN_{7,7,6,4,3,4,6,7}(x), \ x, y \in R^7$$

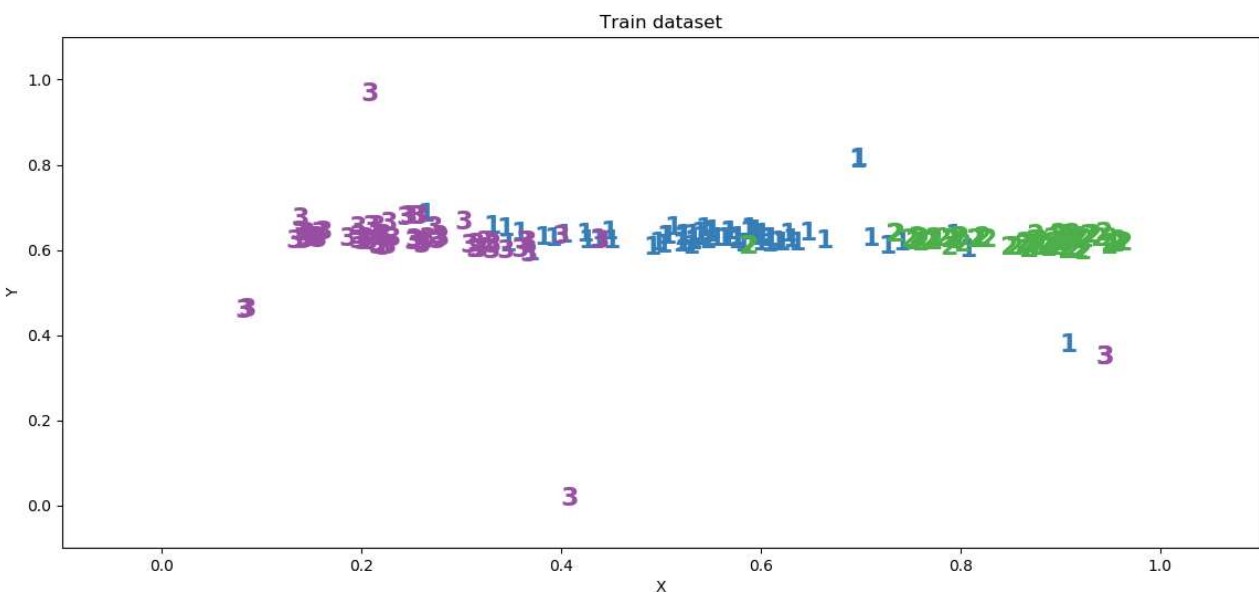

**Figure 11.** Wheat Seeds training dataset (different classes colored differently for convenience).

The ADAM learning algorithm is used for training all auto-encoders. The learning rate is set to $\eta = 0.001$. Each auto-encoder was trained for $3 * 10^5$ epochs.

First, the quality of the DNN neural network classifier and the EnAE classifier were compared according to traditional quality criteria. The calculated values of precision, recall, F-score are presented in Table 9.

**Table 9.** DNN and EnAE quality assessment by traditional criteria.

| Classifier | Precision | Recall | F-Score |
|---|---|---|---|
| DNN | 1.00 | 1.00 | 1.00 |
| EnAE | 1.00 | 0.98 | 0.99 |

The quality of both classifiers is quite high, while DNN slightly outperforms EnAE.

Let us analyze the classifiers using the multidimensional EDCAP criterion. The characteristics of the EDCAP will show the magnitude of the actual class recognition areas relative to the training data in a multidimensional feature space. In the algorithm for calculating the characteristics of EDCAP, the grid spacing of the feature space is chosen to be 0.08. The number of feature space splitting cells was 35,831,808. The calculated values of the generalized EDCAP characteristics for both classifiers are presented in Table 10.

**Table 10.** DNN and EnAE quality assessment by EDCAP.

| Classifier | Excess | Deficit | Coating | Approx | Pref |
|---|---|---|---|---|---|
| DNN | 735,998.84 | 0.10 | 0.90 | 0.00 | 0.00 |
| EnAE | 410.80 | 0.70 | 0.30 | 0.06 | 0.00 |

Comparison of the values of the Excess characteristics for the studied classifiers shows that the recognition area of the DNN classifier outside the training area is significantly higher than that of EnAE. Thus, it can be assumed that the DNN classifier is more susceptible to adversarial attacks than EnAE.

To test the actual resistance of classifiers to adversarial attacks using the algorithm presented in Section 4.1, 1960 modified examples were generated. As a result of classifying these examples, the number of successful adversarial attacks on each classifier was calculated. The results of the experiment are shown in Table 11.

**Table 11.** Number of successful adversarial attacks on classifiers.

| Classifier | Number of Adversarial Attacks |
|------------|-------------------------------|
| DNN | 759 |
| EnAE | 43 |

The data presented in the table shows that more than 17 times more adversarial attacks were successfully carried out against the DNN classifier than against the EnAE classifier. For clarity, Figure 12 shows the training set and successful adversarial attacks on the DNN classifier, and Figure 13—on the EnAE classifier. Class labels that are not recognized in the same way as their neighboring examples of the training dataset are marked in red.

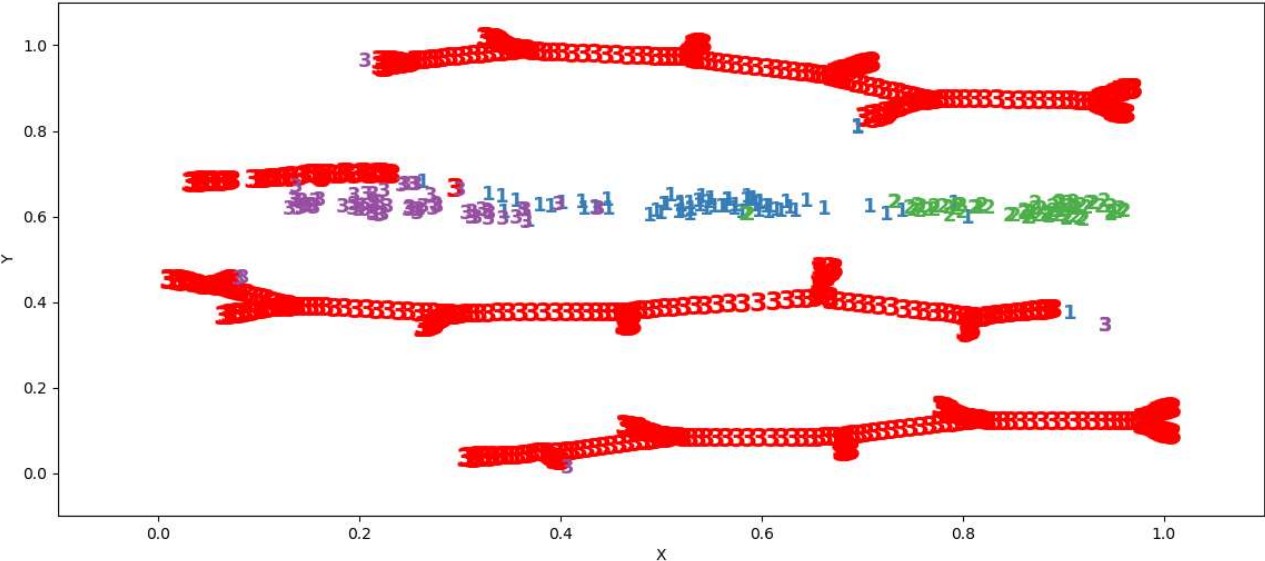

**Figure 12.** Successful adversarial attacks on the DNN classifier (highlighted in red).

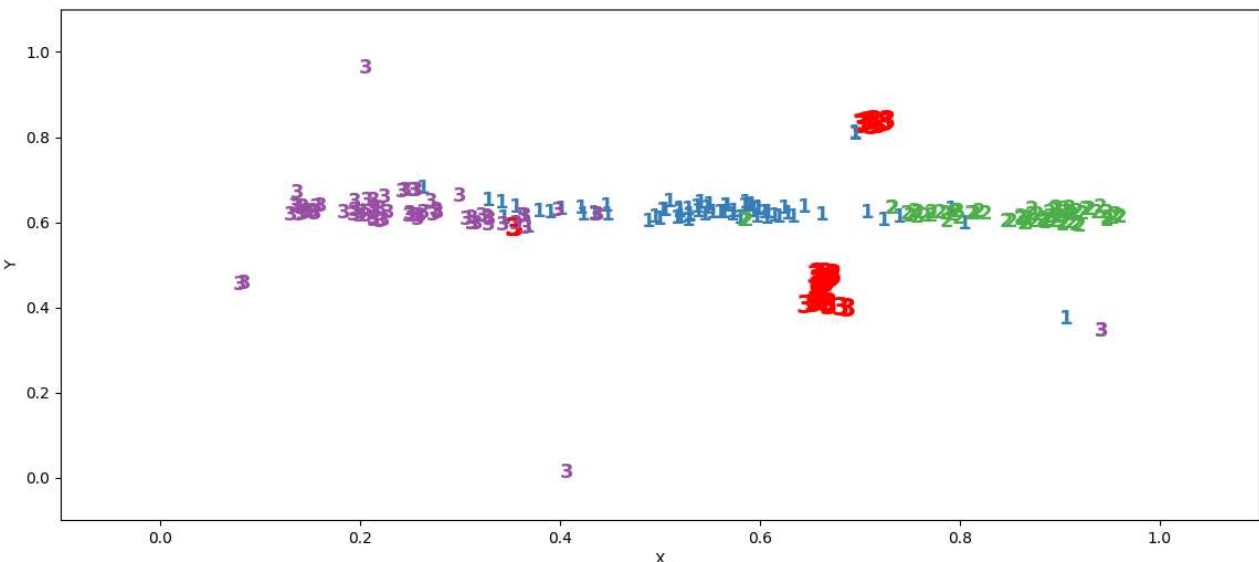

**Figure 13.** Successful adversarial attacks on the EnAE classifier (highlighted in red).

Comparison of the test results and the calculated characteristics of EDCAP makes it possible to verify that, in general, the hypothesis of greater resistance to adversarial attacks of the EnAE classifier is confirmed. It can also be noted that the multidimensional EDCAP criterion makes it possible to characterize the quality of the classifier not only in the area of the training sample, but also beyond it.

To evaluate the effectiveness of the proposed synthesis method, the results of the classification of adversarial examples were evaluated using traditional classification quality metrics. The estimates were calculated on a set of modified examples marked with class labels from the training data set. The evaluation results for DNN and EnAE are shown in Table 12.

**Table 12.** Results of assessing the quality of the classification of adversarial examples.

| Classifier | Precision | Recall | F-Score |
|:---:|:---:|:---:|:---:|
| DNN | 1.00 | 0.61 | 0.61 |
| EnAE | 1.00 | 0.98 | 0.99 |

The values of the traditional quality characteristics are confirmed by the fact that this DNN classifier is more vulnerable to adversarial attacks than the ensemble of auto-encoders (EnAE) synthesized by the proposed method.

The results of the experiments confirm the effectiveness of using EDCAP characteristics to assess the adversarial stability of classifiers.

## 5. Discussion

The main results of this work include:

- a method for investigating the properties of classifiers outside the training sample;
- investigation and identification of the reasons for classifier errors based on machine learning, including adversarial attacks;
- new EDCAP multidimensional criterion for assessing the quality of the multiclass classifier;
- a new EnAE classification method based on ensemble of auto-encoders designed to better resist adversarial attacks than traditional classifiers based on artificial neural networks, SVMs and other approaches.

The key value of this work lies in explaining the reasons for the errors of classifiers and the success of adversarial attacks on them. The proposed method for reducing classifier errors is based on controlling the areas that the classifier recognizes after training. New criteria for assessing the quality of single-class and multiclass classifiers are proposed.

The proposed EDCAP multidimensional criterion makes it possible to compare classifiers according to their level of resistance to adversarial attacks. The criterion is universal and independent of the implementation of the classifier and the strategy for generating adversarial attacks.

The developed synthesis method and EnAE multiclass classification algorithm based on ensemble of auto-encoders in experiments confirms a higher resistance to adversarial attacks than classifiers based on SVM, ensemble methods and deep neural networks. This property is easily explained from the proposed explanation of the nature of adversarial attacks and the properties of auto-encoders.

Note that in the proposed synthesis method, instead of an ensemble of auto-encoders, an ensemble of one-class classifiers of any type, for example, SVM, can be used. It is important that the classification rules and tools for regulating the quality of model training are implemented for the ensemble.

We note the advantages and disadvantages of the proposed methods. The main properties of the proposed multidimensional criteria for assessing adversarial stability are as follows:

- makes it possible to assess the vulnerability of classifier to adversarial attacks and errors before exploitation;
- works for any classifier;
- works in multidimensional space when visualization is difficult;
- evaluation of a multiclass classifier takes into account the intersection of the domains of class definitions and the preferences of the classifier in the area of intersection;

- allows to objectify the quality of classification outside the training sample and to numerically estimate the stability of the actual recognition area of the classifier in a multidimensional space.

The EDCAP multidimensional criterion has the following disadvantages:

- sensitivity to the choice of the size of the feature space partition cell;
- high requirements for RAM for calculations in multidimensional space;
- polynomial complexity of calculations with a degree of a polynomial equal to the dimension of the feature space, which makes it impossible to directly apply the criterion for problems with a large dimension, in particular, in image analysis.

Note that, despite the above disadvantages, alternative methods for evaluating the properties of the behavior of classifiers outside the training set were not found. At the same time, it seems possible to optimize the EDCAP calculation procedure using algorithmic approaches and hardware (GPGPU).

The task of choosing the step of discrete partitioning of the feature space in the working area of the classification seems to be very important and has not been solved yet. The heuristics used by the authors require study.

On the one hand, a too large grid step reduces the computational load. On the other hand, the accuracy of the EDCAP characteristics is reduced, making them useless. Reducing the grid step is also not desirable—the computational load for EDCAP calculations is growing. But, starting from a certain step, a significant number of grid cells in the training data area turn out to be empty, which also makes the numerical values of the EDCAP characteristics unreliable. Obviously, there is some optimal step value, which can be characterized by a heuristic rule: in the definition area of the training set of the class, each cell must contain at least one sample data. Further research may lead to a methodology for calculating the optimal step for a particular data set.

The proposed method for the synthesis of stable EnAE classifiers is characterized by the following properties:

- allows us to synthesize classifiers that are controllably resistant to adversarial attacks and errors;
- properties of protection against adversarial attacks are justified by the properties of the applied auto-encoders;
- it is possible to optimize the synthesis by customizing auto-encoders for each of the classes;
- it is possible to use auto-encoders both based on SVM and based on neural networks.

The EnAE synthesis method has the following limitations:

- the adversarial stability of the classifier is very sensitive to the initial distribution of weight coefficients and the chosen architecture of neural networks of auto-encoders;
- for cases where the number of classes is very large, it takes a lot of resources to prepare a large number of one-class classifiers for each class.

The use of the proposed synthesis method will make it possible to control the level of classifier errors and increase confidence in machine learning methods. Increasing the reliability of machine learning methods will expand the scope of their application in practical applications.

## 6. Conclusions

This paper explores the error problems of classifiers based on machine learning and explains the reasons for adversarial attacks on classifiers. The new multidimensional criterion (EDCAP) for assessing the resistance of multiclass classifiers to adversarial attacks is proposed. The EnAE classification algorithm based on ensemble of neural network auto-encoders is developed and classifier synthesis method is proposed. The EnAE classification algorithm is based on the formation of compact recognition areas for individual classes in the feature space and implements hard voting approach. This feature allows the classifier to differentiate examples of known classes from others that are outside the

area of training samples. Also, it is possible to detect anomaly inputs, which are far from regions of training samples. The multidimensional criterion (EDCAP) makes it possible to evaluate the correspondence of the actual recognition area to the training data and tune the classifier to the required level of resistance to adversarial attacks. The synthesis of two EnAE classifiers for publicly available multidimensional datasets is demonstrated and compared with traditional classifiers based on SVM, soft voting classifier and deep neural networks. The results of the experiments showed the effectiveness of the proposed EnAE classification algorithm in comparison with traditional models. The features, advantages and disadvantages of the proposed methods are noted. The tasks for further work are outlined in order to expand the applicability of the results obtained.

**Author Contributions:** Conceptualization, methodology, A.G. and V.E.; software, validation, visualization A.G.; writing—original draft preparation, A.G.; writing—review and editing, A.G. and V.E.; supervision, V.E.; funding acquisition, A.G. and V.E. All authors have read and agreed to the published version of the manuscript.

**Funding:** The reported study was funded by RFBR according to the research project No. 20-37-90073.

**Institutional Review Board Statement:** Not applicable.

**Informed Consent Statement:** Not applicable.

**Data Availability Statement:** MDPI Research Data Policies.

**Conflicts of Interest:** The authors declare no conflict of interest.

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
