# Peer review of "Quality Criteria and Method of Synthesis for Adversarial Attack-Resistant Classifiers"

_make, doi:10.3390/make4020024_

Round 1

Reviewer 1 Report

This paper proposed an ensemble of quality-controlled one-class classifiers scheme for analyzing adversarial attacks on machine learning based on using the multidimensional EDCAP criterion (Excess, Deficit, Coating, Approx, Pref), and experiments showed that the effectiveness of the proposed EnAE classification algorithm in comparison with traditional models, improves the resistance to adversarial attacks.

Overall, the paper is well-organized and easy to follow.

I like the following part of the paper:

1) clear explanation about the EDCAP criteria;

2) a detailed description of the subphases in the evaluation of the correspondence of the actual recognition area to the training data and tune the classifier to the required level of resistance to adversarial attacks;

3) accuracy results comparison between different approachesbased on SVM and deep neural networks.

One potential improvement I can think of is to also create clear for informative image-based figures (figs 4-11).

Author Response

Thank you very much for the flattering review!

Point 1:

One potential improvement I can think of is to also create clear for informative image-based figures (figs 4-11).

Response 1:

As far as we have understood your remark we updated several figures (1, 5, 6) to provide clear shapes and higher resolution. For other figures (7-13) we considered that some clutter of digits on them emphasizes the combination of traning and adversarial samples better than colored dots.

Reviewer 2 Report

The authors focus on the analysis of adversarial attacks. They also provide a relevant ensemble autoencoder. The overall work is interesting, and the analysis is described well. Some improvement points are given below.

1. In the introduction section, the contributions of this work should be clearly enumerated and described.

2. The authors should include a new section describing relevant works and highlighting their contributions.

3. In general, the resolution of the figures should be improved.

4. In section 3, the authors can include a table explaining all the symbols.

5. In section 3, each formula can be further discussed and analysed. Also, relevant references should be provided. 

6. The evaluation results are poor. For instance, the authors do not provide a comparison with other ensemble algorithms. Therefore, a relevant comparison should be provided.

Author Response

Point 1:

In the introduction section, the contributions of this work should be clearly enumerated and described.

Response 1:

We added the list of our contributions to the end of Introduction.

Point 2:

The authors should include a new section describing relevant works and highlighting their contributions.

Response 2:

We provided Review and Analysis subsections in the Background section with references to relevant works and added special Motivation subsection to highlight the problems have been posed. Also reference [38] to the paper about voting classifiers is appended.

Point 3:

In general, the resolution of the figures should be improved.

Response 3:

We imporved resolutions of figures with blurry shapes (exactly, figure 1, 5, 6).

Point 4:

In section 3, the authors can include a table explaining all the symbols.

Response 4:

We added section 3.1 with description of symbols (Table 1) used further in this section.

Point 5:

In section 3, each formula can be further discussed and analysed. Also, relevant references should be provided. 

Response 5:

In the section 3 every formula has a textual description which preceded it and key formulas are discussed after their introduction. Please point exact formula which caused you questions.

Point 6:

The evaluation results are poor. For instance, the authors do not provide a comparison with other ensemble algorithms. Therefore, a relevant comparison should be provided.

Response 6:

We updated section 4.2 with results of evaluation of Voting Classifier. Several tables (5-8) were updated and figure 9 was added also.  Thank you for your point, because it made our research more profound!

Round 2

Reviewer 2 Report

The authors addressed most of the comments. In general, the paper can be further improved based on the directions of the previous review phase.

Author Response

Thank you very much for reviewing!

In new revision we fixed minor typos, updated literature review with one more reference of voting classifier application and highlighted in abstract and conclusion that EnAE allows to detect anomalies in inputs in opposition to known voting approaches.